# Stability and Antibiotic Potency Improvement of Levofloxacin by Producing New Salts with 2,6- and 3,5-Dihydroxybenzoic Acid and Their Comprehensive Structural Study

**DOI:** 10.3390/pharmaceutics15010124

**Published:** 2022-12-29

**Authors:** Ilma Nugrahani, Muhammad Ramadhan Sulaiman, Chiaki Eda, Hidehiro Uekusa, Slamet Ibrahim

**Affiliations:** 1School of Pharmacy, Bandung Institute of Technology, Bandung 40132, Jawa Barat, Indonesia; 2Department of Chemistry, School of Science, Tokyo Institute of Technology, Tokyo 152-8551, Japan; 3Faculty of Pharmacy, Jenderal Ahmad Yani University, UNJANI, Cimahi 40531, Jawa Barat, Indonesia

**Keywords:** antibiotic potency, 2,6-dihydroxybenzoic acid, 3,5-dihydroxybenzoic acid, levofloxacin, salt, stability

## Abstract

Recently, solid-state engineering has become a promising approach to improving the stability and potency of antibiotics. Levofloxacin (LF) is a broad-spectrum fluoroquinolone antibiotic marketed in solid and solution dosage forms. However, this substance forms solid hydrates under ambient conditions and degrades due to lighting, which may change its solid properties and dose. In addition, resistance cases have been reported due to long-time antibiotic usage. This research aims to allow LF to react with antioxidant dihydroxybenzoic acid (DHBA), which has low antimicrobial activity, to produce a more stable compound under water and lighting conditions and improve LF’s potency. The experiment begins with a screening to select potential DHBA isomers that can react with LF and predict the stoichiometric ratio using phase diagrams, which show that 2,6-DHBA and 3,5-DHBA are prospective antioxidants that can react with LF in a (1:1) molar ratio. Multicomponent systems are prepared by dissolving the LF–DHBA mixture in (1:1) ethanol–methanol (95% grade) and evaporating it. Then, the new solid phase formation is confirmed by thermal analysis and powder X-ray diffractometry. Next, infrared spectrophotometry and neutron magnetic resonance analyses are used to identify the LF–DHBA’s interactions. Finally, single-crystal X-ray diffractometry is used to solve the three-dimensional structure of the multicomponent system. We then conduct a hygroscopicity and stability test followed by a lighting and potency test using the microdilution method. Our data reveal that both reactions produce salts, which are named LF-26 and LF-35, respectively. Structurally, LF-26 is found in an anhydrous form with a triclinic crystal packing, while LF-35 is a hemihydrate in a monoclinic system. Afterward, both salts are proven more stable regarding water adsorption and UV lighting than LF. Finally, both multicomponent systems have an approximately two-fold higher antibiotic potency than LF. LF-26 and LF-35 are suitable for further development in solid and liquid dosage formulations, especially LF-35, which has superior stability compared with LF-26.

## 1. Introduction

Over the decades, multicomponent systems have been developed to modulate the physicochemical properties of active pharmaceutical ingredients (API), including fluoroquinolone antibiotics, such as ciprofloxacin [1,2,3] and levofloxacin (LF) [4]. In general, solid-structure development utilizes either ionic or non-ionic interactions or both [5]. Levofloxacin (LF) is preferred over ciprofloxacin because it has a better pharmacokinetic profile and can be taken once daily. This fluoroquinolone antibiotic has a broad spectrum, with activity against Gram-positive and Gram-negative bacteria. The difference between Gram-positive and Gram-negative bacteria is that Gram-positive bacteria possess a rather thick peptidoglycan layer with no outer lipid membrane, whereas Gram-negative bacteria possess a thin peptidoglycan layer and outer lipid membrane. LF is widely used for treating infections, such as pneumonia, kidney infections, prostate infections, and skin infections caused by Gram-positive bacteria [6]. 

In the market, LF is used in its base and HCl-salt form. Both show pseudopolymorphism. They transform from the anhydrous to the hemihydrate form [7,8], which then adsorbs more water to form monohydrates under 75% RH [4]. This hydrate transformation consequently changes the physicochemical properties of the solid drug. Moreover, like ciprofloxacin [9,10], LF is unstable in water and under lighting. Reports have found this compound unstable under light exposure, especially in the solution phase [10,11,12,13]. This instability occurs not only in the aqueous solution phase but also in the solid phase [13,14,15]; it degrades and produces levofloxacin-*N*-oxide [16]. In addition to being marketed as a solid preparation, fluoroquinolone antibiotics are also prepared in solution forms [17,18]. Therefore, the stability of LF under humidity and lighting conditions is an important issue. Moreover, there are several reports about bacterial resistance cases toward these antibiotics since they have been used for a long time [17,18].

Solid multicomponent systems of fluoroquinolone with other drugs [1,2,3,19] and excipients [20] have been reported before. In addition, the combination of LF with phthalimide and caffeic acid has recently been reported to increase potency [21], as have other antibiotic solid multicomponent systems, such as azithromycin with paracetamol [22] and cefixime [23]. Furthermore, we recently found that salt from LF with the antioxidant citric acid can improve stability and increase potency two-fold [24]. 

In this study, we attempt to combine LF with antioxidant dihydroxybenzoic acid (DHBA) isomers, 2,6-DHBA, and 3,5-DHBA, which are selected for several reasons: Firstly, LF–DHBA’s reactions have never been reported. Information on the compound’s three-dimensional structure is essential for understanding the mechanisms of its performance improvement. Secondly, DHBA compounds are metabolites in fruits, nuts, and vegetables; thus, they are naturally abundant. Next, this benzoic acid derivate group has been reported to show low antibiotic potency [25], which is expected to have a synergic effect on LF’s potency. Finally, some reports have stated that DHBA might exert anti-inflammatory activity [26,27]. On the other hand, LF may cause abdominal pain [28], besides inflammation due to gastrointestinal perforation [29]. Hence, DHBA isomers are economical, safe, and can be expected to add benefit to the parent drugs. 

Some DHBA isomers are found on the market; however, in our preliminary trial, only 2,6- and 3,5-DHBA are observed as prospective compounds that can react with LF. They produce stable off-white or clear crystals, while other isomers are found to produce brown-tan color powders, indicating unstable yields. The pKa values of 2,6-DHBA and 3,5-DHBA at 25 °C are 1.30 [30] and 4.04 [31], respectively. By contrast, LF is a zwitterion compound with two pKa values at 25 °C: 5.35 (carboxylic acid) and 6.72 (piperazinyl ring) [24]. Different pH values may facilitate the multicomponent formation between LF and benzoic acid derivates. In addition, p-hydroxybenzoic acid shows inflammatory inhibition activity by activating the estrogen receptor β (Erβ) [32]. 

With a variety of structural isomers, DHBA has been reported to react with several drugs, such as pyrazinamide with 3,5, 2,6, and 2,4-DHBA [33]; carbamazepine with 2,3-DHBA [34]; ubiquinol with 3,4-DHBA [35]; and piroxicam with 2,5-DHBA [36]. Drug-DHBA multicomponent systems have better solid-state properties, such as higher stability and solubility, than a single API. Based on data from prior research, this antioxidant group is expected to overcome LF’s instability and improve its potency. 

Next, we compose a phase diagram to predict the stoichiometric ratio of LF–DHBAs, followed by multicomponent system preparation. Thermal analysis and powder X-ray diffractometry confirm the new solid phases from solvent evaporation. This is followed by infrared spectroscopy and neutron magnetic resonance to observe the new interactions. Finally, single-crystal diffractometry is conducted to solve the three-dimensional structure thoroughly.

Furthermore, we investigate the stability of the new multicomponent solid phases towards lighting and humidity under ambient conditions. Moreover, microbial tests are conducted to investigate antibiotic potency improvement. These findings may enrich the scientific information in the chemistry and pharmaceutical fields to strengthen solid-state arrangement as a strategy for antibiotic development, including the development of LF [19,20,21,22,23,24]. 

## 2. Materials and Methods

### 2.1. Materials 

Levofloxacin-(S)-enantiomer hemihydrate pro analyses, including levofloxacin (LF), 2,6 dihydroxybenzoic acid (2,6-DHBA), 3,5 dihydroxybenzoic acid (3,5-DHBA), sodium chloride (NaCl), potassium dihydrogen phosphate (KH_2_PO_4_), disodium hydrogen phosphate (Na_2_HPO_4_), potassium chloride (KCl), methanol 95%, ethanol 95%, potassium bromide (KBr) pro analysis, and Mueller–Hinton broth were purchased from Merck—Sigma Aldrich (Darmstadt, Germany). Aluminum pans for DSC were purchased from Rigaku (Tokyo, Japan). Furthermore, distilled water, wild cultures of *Escherichia coli*/*E. coli* (ATCC 8939) and *Staphylococcus aureus*/*S. aureus* (ATCC 6538), and aqueous buffer solutions of pH 1.2, 4.5, 6.8, and 7.4 were prepared at the School of Pharmacy, Bandung Institute of Technology (Bandung, Indonesia).

### 2.2. Methods

#### 2.2.1. Stoichiometric Screening Using a Phase Diagram

The physical mixtures (PM) of LF with 2,6-DHBA and 3,5-DHBA were prepared in the molar ratios of 10:0, 8:2, 7:3, 6:4, 5:5, 4:6, 3:7, 2:8, and 0:10, weighed using a digital scale (Fujitsu FSR-A220, Tokyo, Japan), and gently mixed with a small mortar and stamper to the prevent reaction. Afterward, the melting points of each starting compound and mixture were measured using an electrothermal apparatus. A binary phase diagram was then composed by plotting the molar fraction of LF on the x-axis and the melting point on the y-axis.

#### 2.2.2. Sample Preparation

Based on the phase diagram data, the fixed composition of LF and DHBA was dissolved in a mixture of 95% ethanol and methanol– (1:1). Next, an ultrasonicator (WiseClean WUC-D06H, Gangwon-do, Republic of Korea) homogenized the solution until transparent, and it was filtered using Whatman paper No. 1. Then, the multicomponent solid phases were collected under two crystallization conditions. First, fast evaporation was conducted by filling a Petri dish with the multicomponent solution and storing it in a fume hood at room temperature (25 ± 2 °C). Secondly, slow evaporation used an Erlenmeyer tube filled with the solution and was put under open air in ambient conditions (25 ± 2 °C/70–80% RH). The crystals were then observed using a binocular microscope (Olympus CX21, Tokyo, Japan) under 40× magnification, and an iPhone 12 camera (Apple Inc., Cupertino, CA, US) recorded an image of the new solid-state habits. Finally, an appropriate crystal was analyzed using single-crystal diffractometry (SCXRD). All the solid samples were sieved through an 18-mesh sieve before testing.

#### 2.2.3. Solid-State Characterization

After the samples were produced, all the solid samples were characterized using thermal analysis conducted via semi-manual electrothermal analysis and differential scanning calorimetry (DSC) analysis. Next, they were analyzed using powder X-ray diffractometry (PXRD) to confirm the thermal analysis results. 

##### Electrothermal Analysis

A small sample was filled and tapped into one-side-closed capillary tubes for melting point determination using an Electrothermal IA9000 (Staffordshire, UK) instrument. The apparatus was started at 40 °C with a heat rate of 10 °C/min to determine the melting point. The sample behavior/appearance was observed thoroughly from starting temperature until melting or decomposing via the magnified visual glass hole on the electrothermal, elaborated by digital temperature reading.

##### Differential Scanning Calorimetry (DSC) Analysis

A Rigaku Thermo Plus EVO2 DSC8231 (Tokyo, Japan) was used to observe the thermal profile. A 2–3 mg sample was placed in an enclosed aluminum pan and heated at a temperature range between 30 and 350 °C with a heating rate of 10 °C/min and a nitrogen purge of 50 mL/min. An empty aluminum pan was used as a reference.

##### Powder X-ray Diffractometry (PXRD) Measurement

Powder X-ray diffractometry (PXRD) was performed using a Philips PW 1710 BASED system (Tokyo, Japan) using Cu-Kα (λ = 1.5418 Å) at a tube voltage of 40 kV and a tube current of 35 mA. Sample diffraction was measured over 2θ = 3–40° with a 4°/min scanning rate.

#### 2.2.4. Structural Study

After solid characterization, the samples were analyzed using Fourier transform infrared spectroscopy (FTIR), proton nuclear magnetic resonance (H-NMR), and single-crystal X-ray diffractometry (SCXRD) to determine and elucidate the structure three-dimensionally.

##### FTIR Measurement

The powder/crystal samples were mixed with KBr crystal pro-FTIR analysis, compressed into a tablet, and put in the sample holder. The spectra were measured in the wavenumber range between 4000 and 400 cm^−1^ with 4 cm^−1^ resolution using a Jasco FT/IR-4200 type A apparatus (Easton, PA, USA).

##### H-NMR Analysis

The two-dimensional structures of the samples were determined by using an H-NMR NMReady-60 by Nanalysis (Calgary, AB, Canada) at 60 MHz, and sample temperatures were kept at 37 °C. First, LF, DHBAs, and the obtained samples were dissolved in deuterium water (D_2_O), and tetramethyl silane (TMS) was added as an internal standard. The solution was analyzed at 0.179 Hz/pt resolution, and relaxation paused at 1.929335 s. Finally, the spectral calculation was performed using Nanalysis NMReady v2.0.7 (Calgary, AB, Canada).

##### SCXRD Analysis

The appropriate form and size of the multicomponent crystals were observed under a microscope. Next, they were selected and put in the Rigaku R-AXIS RAPID sample holder (Tokyo, Japan). Analysis using radiation of Cu-Kα (λ = 1.5418 Å) was utilized with a graphite monochromator at 173 K. ABSCOR corrected for the absorption effects. The structure was solved using a dual-space algorithm of SHELXT and then refined on F2 using SHELXL-2017/1. The non-hydrogen atoms were refined anisotropically. A differential Fourier map pointed to the hydrogen atoms attached to nitrogen and oxygen atoms except for water oxygen atoms, which were treated using the hydrogen atoms and a riding model at the geometrically calculated position. The water hydrogen atoms were isotropically refined with the standard bond distance restraints. Finally, Mercury (CSD System, version 4.3.1, 2018) was used to graph the molecular structures.

#### 2.2.5. Stability Study

##### Test of Stability towards Lighting and Humidity

The stability test was conducted under high humidity and UV lighting for four weeks of observation. First, stability towards humidity was tested in the controllable environmental chamber EYELA KCL-2000A (Tokyo, Japan), which was set at 75 ± 1.0% RH/25 ± 0.5 °C, representing the actual conditions of Indonesia as a tropical country. First, a 1 g sample was weighed, dispersed in a Petri dish (d = 7.5 cm), and stored in the EYELA chamber for four weeks. Sampling was conducted every day of the first week, then twice a week for the following weeks. Next, the adsorbed water amount was measured by Karl Fischer (KF) titrator Mettler Toledo V20 (Giessen, Germany) with Aquastar Combi-Titrant 5 reagent from Merck-Sigma (Jakarta, Indonesia). Before the analyses, 1 mL of KF reagent was calibrated, equal to 4.7 mg of water. In this test, a 25 mg sample was added to the KF titrator chamber containing the reagent. Then, the analysis started after the drift value was less than 25 mg/min and finished after the water amount was stable [37].

The photostability test was performed using a Labonce-150 TPS photostability chamber (Beijing, China), referring to the protocols recommended in the ICH topic Q1B guideline regarding photostability testing of new active substances and medicinal products [38]. It is performed by exposing ~1 g of sample to 200-Watt h/m^2^ of UV light (320 to 400 nm) and 1.2 M lux h/m^2^ of visible light (400–800 nm) for four weeks under the condition 75 ± 1.0% RH/25 ± 0.5 °C. Afterward, sampling was conducted weekly and subjected to analysis by the Jasco FTIR-4200 type A apparatus (Easton, PA, USA) and UV–visible spectrophotometer HP/Agilent 8453 (Santa Clara, CA, USA) instrument. 

To measure the concentration of stability tests samples, the absorption profiles of samples in aqueous solution were scanned in the 200–400 nm wavelength region using a UV–visible spectrophotometer HP/Agilent 8453 (Santa Clara, CA, USA). The obtained spectra of the multicomponent system solutions were first-order derived and used the spectra zero-crossing points to analyze the LF concentration. Finally, the specificity, linearity, accuracy, precision, limit of detection (LOD), and limit of quantification (LOQ) were determined to fulfill the validation requirements [39].

#### 2.2.6. Antimicrobial Activity Study

The minimum inhibitory concentration (MIC) was determined via the liquid micro-dilution method with triplicate testing based on the Clinical and Laboratory Standards Institute (CLSI-2012) guideline [40] using Mueller–Hinton Broth (MHB) in a 96-well microplate with a 2-fold sample dilution step. The bacteria utilized in this study were non-resistant strains of *S. aureus*-ATCC 6538 (Gram-positive bacteria) and *E. coli*-ATCC 8939 (Gram-negative bacteria). The bacterial suspension was prepared from similar morphology colonies cultured at 37 ± 0.2 °C for 24 h in MHB. Then it was suspended to 0.5 McFarland (1.5 × 10^8^ CFU/mL). Following the guideline, the final density of bacteria suspension should be 2–8 × 10^5^ CFU/mL [40]. Therefore, the density of bacteria in the broth was prepared by diluting 20 µL of bacterial suspension in 80 mL broth. 

The antimicrobial activity study was performed by dissolving samples in buffer solutions pH 6.8 and pH 7.4, which were prepared as follows [39]:-The phosphate buffer solution pH 6.8 was made from 13.872 g of potassium dihydrogen phosphate (KH_2_PO_4_) and 35.084 g of disodium hydrogen phosphate (Na_2_HPO_4_) and dissolved in distilled water to produce a 1000 mL buffer solution. -The phosphate-buffered saline solution pH 7.4 was prepared by dissolving 8.0 g of NaCl in 800 mL of distilled water in a 1000 mL volumetric flask. Afterward, 0.2 g of potassium chloride (KCl), 1.44 g of Na_2_HPO_4_, and 0.245 g of KH₂PO₄ were added to the solution, which was then adjusted to 1000 mL. -Distilled water was boiled for 15 min to evaporate all CO_2_ and measured as ~ pH 7.0 medium.-The pH value of each solution was determined using a pH meter (Mettler Toledo, Darmstadt, Germany).

The microdilution was done in 96 wells, with 12 rows and 8 columns, and all samples were observed in triplicate. In detail, we filled the wells in the first row (row number 1) with MBH without any bacteria and sample solution, which was the negative control. Next, wells in the second row were only filled with bacteria and MBH solution, named positive control. The starting concentration of the sample was prepared at 256 µg/mL in the tested buffer solutions. For the dilution step, the starting sample solution was mixed with MHB in a 1:1 ratio, which means that the highest concentration was 128 µg/mL (12th row). Each well in the row contained half of the previous wells’ samples. For each well, 10 µL of bacterial suspension was added. The 2-fold dilution in 10 steps provided the lowest concentration of 0.0625 µg/mL (3rd row). The final density of bacteria in the mixture became approximately 3 × 10^5^ CFU/mL. 

All the microdilution trays containing the samples and controls were incubated at 37 °C for 24 h. Next, the presence or absence of bacterial growth was compared against the controls. The MIC value was determined using an Insten Magnifying Glass 10× Handled (Hereford, UK), as the lowest concentration did not show bacterial growth. 

Finally, the combined effects of levofloxacin–DHBA were checked using the checkerboard method based on the Clinical Microbiology Procedures Handbook 3rd edition [41]. The combined effect was calculated using the fractional inhibitory concentration index (FICI) equation. ΣFICI value represented the synergy effect ≤0.5, while the indifferent and antagonist results were indicated by ΣFICI value >0.5 to <2 and ΣFICI value ≥2, respectively [42]. 

#### 2.2.7. Statistics

All the experiments were performed in independent triplicate trials. The quantitative analysis data average and deviation standard calculation, curve making, and graph composition were conducted using Microsoft Excel (Microsoft Corp., Redmond, WA, USA) via the Student’s t-/ANOVA test method. Mercury (CSD System, version 4.3.1, 2018) was used to calculate and draw the structures from SCXRD data.

## 3. Results and Discussion

Firstly, a series of screenings were performed to find the DHBA isomers which can interact with LF, 2,4-, 2,6-, 3,5-, and 3,6- DHBA. However, only 2,6-DHBA and 3,5-DHBA were promising compounds to interact with LF, and the others produced unstable compounds due to the oxidation reaction observed. Based on the “W” profile of the phase diagrams in Figure 1, the formation of the LF–DHBA multicomponent system occurred in the (1:1) molar ratio, which is shown by the melting point between the two eutectic points. This stoichiometric ratio was also found in the reaction of LF with other organic compounds, namely, phthalimide [21], caffeic acid [21], and citric acid [24].

The evaporation of the (1:1) molar ratio of LF–DHBA in the ethanol–methanol 95% grade solution produced a needle-like form of LF-26 and a cubic form of LF-35, in a clear to an off-white color, as depicted in Figure 2. Electrothermal measurement showed that the new multicomponent systems had higher melting point than their starting materials, indicating that they were new solid phases named LF-26 at 254 °C and LF-35 at 282 °C, respectively. 

Figure 3 depicts the DSC thermogram of the starting materials and their combination compounds. LF’s thermogram at the bottom of the figure shows endothermic peaks at 117 °C and 225 °C. This profile indicates a hemihydrate form of LF; the first curve represents the breaking point of the water molecules, while the second curve is LF’s melting point [24]. The thermogram demonstrated 166 °C as the melting point of 2,6-DHBA and 237 °C as that of 3,5-DHBA, which agrees with past research [43,44], confirming their identity and purity. The thermograms also provided the necessary data for the multicomponent systems, revealing the melting point of LF-26 at 254 °C and LF-35 at 282 °C, which was in line with the electrothermal measurement data. Moreover, LF-35’s thermogram also shows a thin endothermic slope, representing water molecule release during heating at ~50 °C, showing that this salt is a hydrate. All multicomponent system’s thermal profiles ended with an exothermic peak, indicating decomposition.

Moreover, Figure 4a,b show that LF-26 and LF-35 have different diffractograms from their PM and starting materials, indicating that new solid phases were composed. The calculated X-ray diffraction patterns were compared to the experimental results from the measurement of the new multicomponent systems, as depicted in Figure 4c for LF-26 and Figure 4d for LF-35, which show their similarity. The calculated diffractograms were collected from our registered structure in the Cambridge Crystallographic Data Centre (CCDC), numbered 2180214 and 2180221 for LF-26 and LF-35, respectively. 

To give further detail, the experimental diffractogram of LF-26 matched the calculated profile with distinctive peaks at 2θ = 6.6; 7.2; 8.4; 13.2; 14.3; 15.4; 17.5; and 20.2° (Figure 4c), while the measured and calculated diffractograms of LF-35 in Figure 4d had peaks at 2θ = 8.9; 10.6; 12.3; 12.4; 14.6; 17.6; 18.3; 19.9; and 21.6°. Thus, all the experimental diffractograms confirmed LF-26 and LF-35 as new solid phases [45,46,47].

The structural study began with an FTIR measurement resulting in spectra in Figure 5. The infrared spectra of the starting materials and PM of LF-2,6-DHBA in Figure 5a show O-H stretching vibrations at 3493 cm^−1^, indicating a hydrate form of LF [24]. Furthermore, the new multicomponent, LF-26, shows a band at 2507 cm^−1^ that is attributed to the nitrogen atom’s protonation from the piperazinyl ring of LF [24,48], which composed the ionic bond in LF-26. In addition, spectra at 1619 and 1346 cm^−1^ correlated with the asymmetric and symmetric O-C-O stretches, respectively. These data indicate proton transfer from the -COOH of 2,6-DHBA, suggesting that the multicomponent LF-26 is a salt form. 

The spectra of LF-35 compared to its PM and starting materials are depicted in Figure 5b. The O-H bond shows a sharp medium-intensity band at 3432 cm^−1^, indicating the water crystal’s hydrogen bonding appearance in the new multicomponent system. Moreover, a small spectrum element is apparent at 2507 cm^−1^, the same as that of LF-26, representing the nitrogen atom’s protonation from the piperazinyl ring of LF [48,49], while the C = O stretching vibration is shown at 1700 cm^−1^ [50]. In addition, there are spectra at 1569 and 1373 cm^−1^ due to asymmetric and symmetric O–C–O stretches, respectively, which suggest proton transfer from the –COOH group [24].

All the FTIR data confirmed that both LF-26 and LF-35 are ionic compounds due to proton transfer from the –COOH group of benzoic acids. Furthermore, LF-35 is predicted to have water molecules in its solid structure or to form a hydrate, which must be confirmed by SCXRD analysis.

Regarding the structural study, ^1^H-NMR observed the hydrogen-1 (^1^H) nuclei position and checked the stability of the multicomponent systems in the solution state. Figure 6 compiles all the ^1^H-NMR spectra, and Figure 7 elaborates on these data by including the numbering of the molecule structure compilations. 

Figure 6a shows that LF’s spectra produced signals in the range δ: 1.8–7.8 ppm, which detected eight types of protons. First, as revealed by the data in Figure 7a, the singlet signals at δ: 7.8, 7.6, and 7.3 ppm refer to the 5-H proton or aromatic group due to the coupling with the F atom in position 6. Next, a sharp singlet signal at δ: 4.5 ppm represents the 2-H proton in positions 2, 3, 5, and 6. Then, the duplet signal at δ: 3.5 ppm describes the 1b position of the 2-H protons, and δ: 2.7 and 2.4 ppm are attributed to 4a and 1c methyl groups, respectively. Finally, at δ: 1.8 ppm, the duplet signals may reflect 3a carboxyl groups [20,51]. 

Regarding 2,6-DHBA, Figure 6b depicts the spectral signals at δ: 6.8–6.4 ppm. First is the duplet signal at δ: 6.8 ppm, which may correspond to Figure 7b as the aromatic protons in positions 3 and 5; the second is a single signal at δ: 6.4 ppm, which shows the aromatic proton in position 4 [43]. 

Figure 6c reveals the spectra of 3,5-DHBA to have δ: 3.5–7.3 ppm, indicating three types of protons. The 1H-NMR spectra showed a doublet at δ: 7.3 and a triplet at δ: 6.4 ppm, which, as shown by Figure 7c, are the aromatic protons in positions 2 and 6. The meta coupling occurred at the proton positions 4 to 2 and 6 [44]. A singlet signal at δ: 3.5 ppm may be attributed to the residual water molecule. 

Figure 6d,e reveals the NMR spectra of the two multicomponent systems, with the atom’s position clarified in Figure 7d,e. LF-26 has five distinctive signals representing five types of protons in the range of δ: 1.5–8.4 ppm. The singlet signal at δ: 8.4 ppm refers to the aromatic group coupled with the F atom in position 6. Moreover, singlet signals at δ: 5.9, 3.5, and 2.9 ppm are attributed to the 1-H of the 2-alkene, 4′amino group, and 5-alkene, respectively. Finally, the carboxyl groups of the 1, 2, 6, and 3a positions were observed at δ: 1.5 ppm. 

The 1H-NMR spectra of LF-35 showed peaks of δ: 1.5–8.6 ppm consisting of six distinctive signals representing six types of protons. Like LF-26, the singlet signal at δ: 8.6 ppm means the protons from the aromatic group are coupled with the F atom in position 6. The singlet signals at δ: 6.8, 6.3, 3.5, 2.9, and 1.5 ppm reflect 1-H in positions 2 and 6; 1-H in position 4; 1-H in the 4′amino group; 1-H in the 5-alkene group; and the carboxyl groups in positions 1, 3, 5, and 3a, respectively. The stability of the salt structure was deemed suitable since it exhibited no breakage of the multicomponent compounds in the solution state.

An SCXRD measurement was performed to finalize the structural study of the new compounds three-dimensionally. The crystallographic data are listed in Table 1, and the structure image can be seen in Figure 8 (LF-26) and Figure 9 (LF-35). Table 1 shows that LF-26 (LF:2,6-DHBA = 1:1) crystallized in the triclinic *P*1 space group with two LF cations and two anions of the BA in the asymmetric unit (Z = 2). Simultaneously, LF-35 (LF: 3,5-HBA: H_2_O = 2:2:1 = 1:1:0.5) crystallized in the monoclinic *P*2_1_ space group with two LF cations and two benzoic acid anions as well as one water molecule in the asymmetric unit.

Figure 8a reveals the proton transfer in LF-26, which was confirmed by the geometry of the COO–group of 2,6-DHBA (shorter C–O were 1.249 and 1.260 Å, and longer C–O were 1.290 and 1.293 Å). Data showed that the difference between the two C–O distances was smaller than that of the COOH group, confirming a salt formation. An intermolecular NH+…O–type ionic interaction (charge-assisted hydrogen bond) was observed, and such an interaction in the crystal increases the lattice energy of the drug crystal. Moreover, intramolecular hydrogen bonds occurred in both LF and DHBA. 

Figure 9a shows that LF-35 was also a salt form. Both DHBA isomers used were in anion form with COO– having almost similar C–O bond lengths (1.267 to 1.269 Å) due to resonance stabilization in the carboxylate anion [52,53]. Because the multicomponent systems were salts, hereafter, LF-26 can be referred to as levofloxacin 2,6 dihydroxybenzoate, while LF-35 represents levofloxacin 3,5 dihydroxybenzoate.

Figure 9c depicts the packing viewed along the *b*-axis, showing a layered structure in 3,5-DHBA (green and magenta) and LF (red). A similar layered structure was also observed in LF-26, as shown in Figure 8d. In both LF-26 and LF-35, the quinolone moieties of LF were held together by π-π interactions. Similar packing arrangements also occurred in several multicomponent crystals of a fluoroquinolone structure, ciprofloxacin, with other benzoic acid derivatives, 4-hydroxybenzoic acid, 4-aminobenzoic acid, and gallic acid [53]. 

There were also water molecules within the crystal lattice of LF-35, which relates to its thermogram in Figure 3. Observing the *c*-axis in Figure 9d, the 3,5-DHBA molecules exist in two different orientations, forming a crisscross pattern within the lattice. This layer was predicted to provide a barrier for the LF molecules, preventing LF degradation from oxidation and thus modulating its stability against UV light [20].

In addition, it is known that the LF-26 crystal has two sets of independent molecules (Z′ = 2). This may appear as a *P*1 structure, but the *P*1 structure is correct because both LF molecules are S-body (chiral compound), and the crystal structure should not include an inversion center. Moreover, LF-35 may appear as a *P*2_1_/c structure, but the *P*2_1_ structure is correct for the same reason. These structures also contain a disorder around the chiral carbon. The methyl group attached to the chiral carbon atoms has equatorial or axial conformation disorder. After determining the novel multicomponent system’s structures thoroughly, we tested for physical stability towards hydrate transformation and chemical resistance against UV lighting. Antibiotic potency tests were then conducted using a common microdilution method [54], which obtained data as follows. 

Figure 10 depicts a hydrate transformation diagram of the LF and LF–DHBA salts, which reveals that the water content of LF increased significantly after three days of storage under the 75 ± 1.0% RH /25 ± 0.5 °C condition. Initially, LF was confirmed by PXRD and DSC analyses to be a hemihydrate and contained ~2.5% water. However, after 3 days, it transformed into a monohydrate form with more than 5% water content. This phenomenon is in line with previous reports, which noted that LF easily changes to its higher hydrate form [4,7,55]. The increasing water content may be due to the LF’s spacious crystal packing and hydrogen bonding capacity, which supports the interaction between this antibiotic and water molecules. The water molecule uptake in the LF hemihydrate caused the hydrate form transformation, consequently changing the solid properties, such as solubility, stability, hygroscopicity, etc. [56]. Recently, LF has been commonly used in its salt form, levofloxacin hydrochloride/LF-HCl. However, LF-HCl also shows a tendency toward hydrate transformation [8], such as that of the LF base, which can change the solid characteristics and dose [14].

By contrast, LF-26 was steady in its anhydrous form (~0.5% water), and LF-35 was steady in its hemihydrate form (~2.2% water). This result was also attained in a study on levofloxacin–citric acid salt, in which the water molecule of LF was replaced by a citric acid molecule on the N-methylamine site [24]. The data show that the salts absorbed a lesser amount of water molecules than LF alone, probably due to the surface property change and the more compact packing of the salts compared with that of the parent compound [56].

The data for the test of stability towards lighting are shown in Table 2, depicting the samples’ visual appearances for four weeks. LF changed from off-white to a more intense yellow, while the appearance of LF-26 and LF-35 remained relatively unchanged. Bofill et al. observed a similar phenomenon in their research regarding ubiquinol multicomponent systems with DHBA, which showed high stability to oxidation indicated by the unchanging color of the multicomponent systems even under stress for 435 days [35]. 

According to previous reports, one of the degradation pathways of LF alters the N-methyl piperazine moiety, specifically by the oxidation process and by opening the ring of the piperazine moiety [14,16]. 2,6-DHBA and 3,5-DHBA have antioxidant activity, as detected by ABTS (2,2′-azino-bis(3-ethyl benzothiazoline-6-sulphonic acid), FRAP (ferric reducing antioxidant power), and CUPRAC (cupric ion reducing antioxidant capacity) assays [27,57]. Furthermore, based on the structural data, the DHBA moiety of the multicomponent system, which sits on a layer, may prevent UV light from passing through and thus deter the alteration of LF’s N-methyl piperazine moiety, giving stability toward the oxidation process [24,58].

Furthermore, the stability profile was confirmed by a concentration measurement using the validated UV–visible spectrophotometer HP/Agilent 8453 (Santa Clara, California, USA) at λ = 288 nm [59], in which 2,6-DHBA and 3,5-DHBA did not interfere with the spectra of LF [60,61]. This method fulfilled the validation parameters with the R^2^ value of the calibration curve = 0.995. In the stability analysis, the level of LF was calculated after reducing the water content, which was determined by KF titration. Hereafter, only the change to the LF is presented.

Stability test results are depicted in Table 3 and Table 4. Table 3 shows that all the samples were relatively stable under the protected storage.

Table 4 reveals that LF concentration in its single preparation dropped ~6.0% after the testing period for stability towards UV lighting, initially 99.9% *w*/*w* falling to 93.9% *w*/*w*. By contrast, LF-26 and LF-35 retained their LF concentrations for four weeks under the test conditions. Notably, the LF concentration of LF-35 was remarkably consistent at 99.9% after four weeks, while the LF concentration of LF-26 was marginally better than its single compound, only dropping 1.3%, from 99.8% to 98.5%. Thus, these results are in agreement with the visual appearance observations. 

Next, Figure 11 shows the degradation curves of LF compared to LF-26 and LF-35, which depicts less steep degradation curves for LF-26 and LF-35 than for LF alone. 

In line with previous reports, LF adsorbed water changed to its higher hydrate [4] and became unstable under light exposure, which is represented by its color change from pale to burning yellow and confirmed by the content decrease [14,15,16]. These results are similar to levofloxacin-citrate (LC), which nevertheless exhibited better stability than LF alone [24]. Like citric acid molecules, both DHBAs protected LF by interacting with the N3 of the N-methyl piperazine site. Moreover, they can prevent water adsorption into the crystal lattice. However, based on the data, LF-35 had superior stability towards lighting than LF-26 and LF alone.

We also measured the antibiotic potency of the new multicomponent systems compared to LF and PM by determining the minimum inhibitory concentration (MIC). LF’s antimicrobial activity has been widely reported to be influenced by its environment’s pH value [24]. Therefore, the microdilution method was performed to determine the minimum inhibitory concentration (MIC) in media with pH values of 6.8 and 7.4 to simulate the gastrointestinal and plasma pH values, respectively [24,62,63]. Previously, the sterility and fertility of the medium for the bacteria used showed that these media were appropriate for the test. 

The MIC data from the sample concentration range of 0.0625–64 µg/mL are shown in Table 5 and Figure 12. In the media pH 6.8 and 7.4, the MIC values of LF against both strains were determined to be 0.125 µg/mL, while the MIC values of LF-26 and LF-35 were the same: 0.0625 µg/mL. These phenomena demonstrated that both new salts reduced the MIC value of LF by two-fold in the pH 6.8 and 7.4 media for the tested strains. The increase in antimicrobial effect may be supported by the synergic work of 2,6-DHBA and 3,5-DHBA with LF, which was confirmed by measuring their FΣFICI [42]. Prior research reported that both compounds have a minor potency toward *S. aureus* and *E. coli* with MIC values of 2 mg/mL and 3 mg/mL, respectively [25]. 

Previously, Campos et al. reported that compound DHBAs of the phenolic acid group exhibited antimicrobial activity through their partial lipophilic character. They hypothesized that the DHBAs desaturated protein in the lining of cells was caused by the passing of the undissociated form of phenols through the cell membrane via passive diffusion, thus disrupting cell structure and lowering the pH value of the cytoplasm [64]. This experiment also studied the possibility of antimicrobial interaction between the constituents by testing the physical mixtures (PM) of LF and each DHBA against the tested strains. The results showed that the MIC values of PM in the pH 6.8 and 7.4 media were 0.0625 µg/mL, revealing an antimicrobial interaction between LF and each DHBA.

The potency data from the present experiment revealed that LF-26 and LF-35 increased the potency of LF by ~2-fold, which might occur due to similar synergistic antimicrobial activity between LF and DHBAs. This result is comparable to a previous study using ethyl 3,4-dihydroxybenzoate (EDHB), which showed a MIC decrease against the tested strains [65]. Nevertheless, the exact mechanism of the potency improvement of the multicomponent system still requires further study since it is expected to overcome cases of antimicrobial resistance.

The FICI values of each combination concentration against *S. aureus* and *E. coli* strain in pH 6.8 and 7.4 are listed in Table 6, which were calculated by the formula: ΣFICI = (MIC LF in combination/MIC LF alone) + (MIC DHBA in combination/MIC DHBA). Due to the high MIC of the DHBAs, their FICI value was equal to that of their mixture with LF, which was 0.062. Hence, LF-2,6 DHBA and LF-3,5 DHBA combinations had the same ΣFICI values (0.452) in every buffer solution pH; less than 0.5 indicated that the LF–DHBA combinations had a synergistic effect [42].

To conclude, the composition of LF–DHBA multicomponent systems, especially LF-35, offers advantages, including stability, solubility, and potency improvement. Furthermore, there are pharmacological advantages that may also be investigated. As mentioned in the introduction, LF has been reported to cause a disturbance in the gastrointestinal tract [28,29], and DHBA may be expected to rectify this [30]. Xu et al. [30] suggested that DHBA compounds are potential protectors against intestinal mucosal inflammation due to perforation by LF, thus alleviating abdominal pain symptoms. They also observed the protective role of p-hydroxybenzoic acid against induced mucosal damage to mouse gastrointestinal tracts. Their results showed that p-hydroxybenzoic acid exerted inflammatory inhibition activity by activating the estrogen receptor β (Erβ) [30]. Moreover, numerous studies have reported the biological properties of hydroxybenzoic acids, i.e., anticancer [66], antimicrobial and antioxidant [25,67], chemo-preventive activities [68], and anti-inflammatory activities [69]. In summary, the many advantages of this antibiotic–antioxidant multicomponent system may be further developed and observed. 

In addition, phenolic acids, including DHBA compounds, are abundant in plant-based dietary ingredients. This is a promising aspect for future studies regarding phenolic acids as potential supplements or functional food ingredients to attain significant biological properties and prevent certain diseases. However, some toxicological concerns remain. One consideration is the lack of information about dosage-related effects and pharmacological interactions with conventional drugs. Moreover, numerous biological properties may interfere with a drug’s efficacy when phenolic acids are consumed simultaneously. This factor should be evaluated when phenolic acids and their derivates are used in drug formulation [70]. 

Finally, based on the experimental data, the reactions with 2,6-DHBA and 3,5-DHBA improved LF’s stability and antimicrobial potency; however, LF-35 was superior to its counterpart. Therefore, this salt can be developed further in solid and liquid dosage forms. All experimental results of this study enrich and strengthen prior reports that solid-state engineering is suitable for improving antibiotic potency. 

## 4. Conclusions

From this experiment, two salts of LF with 2,6-DHBA and 3,5-DHBA combinations, named LF-26 (levofloxacin-2,6-dihydroxybenzoate) and LF-35 (levofloxacin-3,5-dihydroxybenzoate), were successfully prepared and their stability and potency were evaluated. Their three-dimensional conformations show that LF-26 is an anhydrate in a triclinic system, while LF-35 is a hemihydrate conformation in monoclinic packing. These antibiotic–antioxidant multicomponent phases were more stable towards water adsorption than LF, which showed pseudopolymorphism under ambient conditions for three days. They also showed superior resistance toward UV than LF. The chemical content of LF hemihydrate decreased by ~6% *w*/*w* after four weeks of lighting; meanwhile, the new salts were stable due to the protecting effect of the DHBAs upon the potentially oxidative-degradable site of LF. Moreover, both salts showed a higher antibiotic potency by ~2-fold than the parent drug. Hereafter, these antibiotic–antioxidant multicomponent systems are suitable for further development in liquid and solid dosage formulation, especially LF-35, the most stable compound compared with LF-26 and LF alone.

## Figures and Tables

**Figure 1 pharmaceutics-15-00124-f001:**
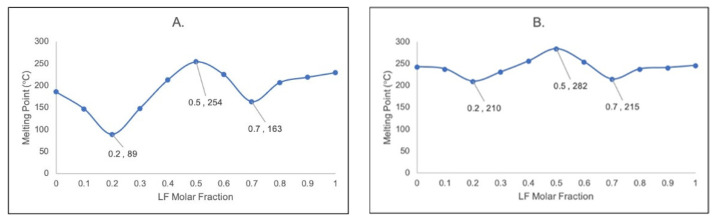
Binary phase diagram of (**A**) LF + 2,6-DHBA; (**B**) LF + 3,5-DHBA. Note: LF = levofloxacin; DHBA = dihydroxybenzoic acid.

**Figure 2 pharmaceutics-15-00124-f002:**
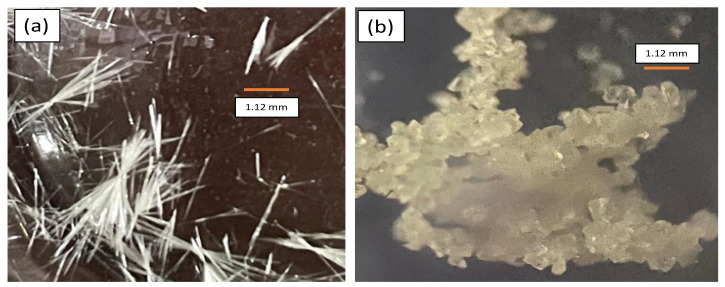
Single-crystal appearance of (**a**) LF-26 and (**b**) LF-35, both under 40x magnification. Note: LF-26 = levofloxacin 2,6 dihydroxybenzoic acid multicomponent system, LF-35 = levofloxacin 3,5 dihydroxybenzoic acid multicomponent system.

**Figure 3 pharmaceutics-15-00124-f003:**
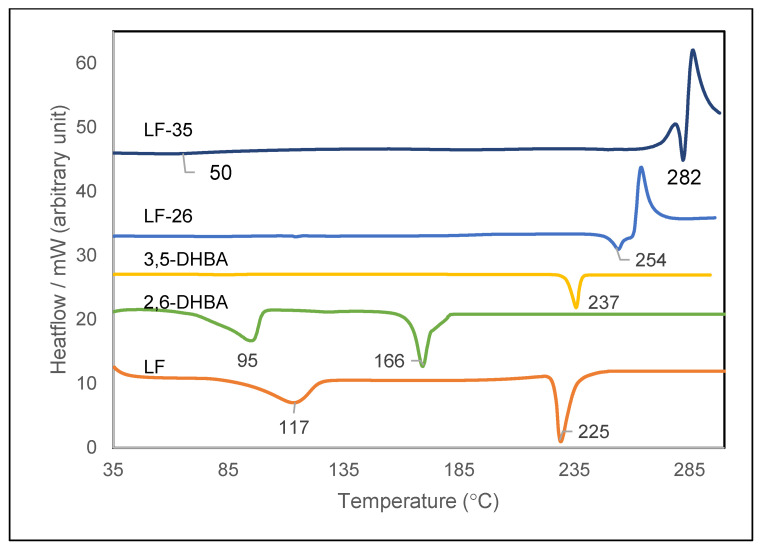
Thermograms of LF-26 and LF-35 from DSC analysis compared to starting materials. Note: LF = levofloxacin (hemihydrate); 2,6 DHBA = 2,6 dihydroxybenzoic acid; 3,5 DHBA = 3,5 dihydroxybenzoic acid; LF-26 = levofloxacin 2,6 dihydroxybenzoic acid multicomponent system, LF-35 = levofloxacin 3,5 dihydroxybenzoic acid multicomponent system.

**Figure 4 pharmaceutics-15-00124-f004:**
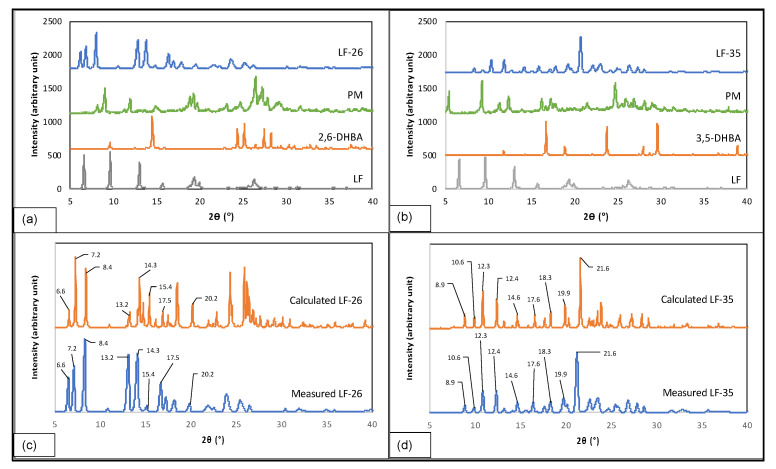
Diffractogram compilation of the multicomponent systems and starting materials of LF-26 (**a**); LF-35 (**b**); as well as the experimental and calculated patterns from single-crystal data of LF-26 (**c**) and LF-35 (**d**). Note: LF = levofloxacin; LF-26 = levofloxacin 2.6 dihydroxybenzoic acid multicomponent system; LF-35 = levofloxacin 3.5 dihydroxybenzoic acid multicomponent system.

**Figure 5 pharmaceutics-15-00124-f005:**
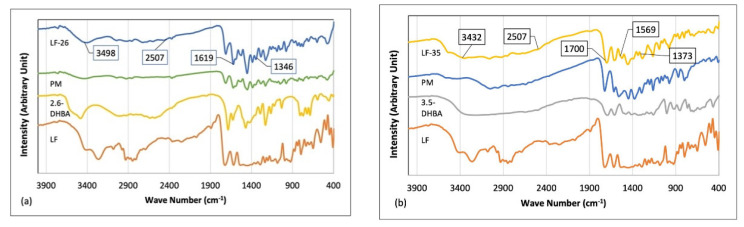
FTIR spectra comparison of the physical mixture and its multicomponent system: (**a**) LF-2,6-DHBA; (**b**) LF-3,5-DHBA. Note: LF = levofloxacin; DHBA = dihydroxybenzoic acid; PM = physical mixture.

**Figure 6 pharmaceutics-15-00124-f006:**
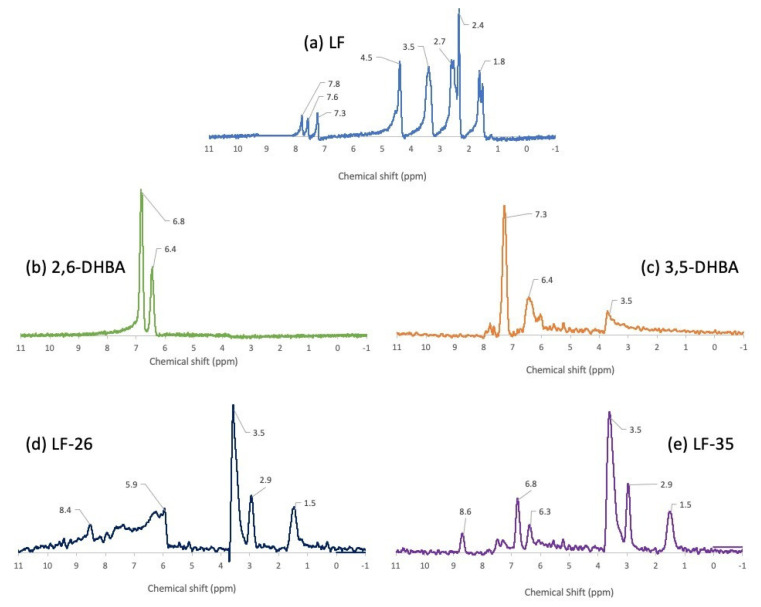
(**a**) ^1^H-NMR spectra of levofloxacin (LF); (**b**) 2,6-dihydroxybenzoic acid (2,6-DHBA); (**c**) 3,5-dihydroxybenzoic acid (3,5-DHBA); (**d**) levofloxacin-2,6-dihydroxybenzoic acid (LF-26); (**e**) levofloxacin-3,5-dihydroxybenzoic acid (LF-35) in deuterium (D_2_O).

**Figure 7 pharmaceutics-15-00124-f007:**
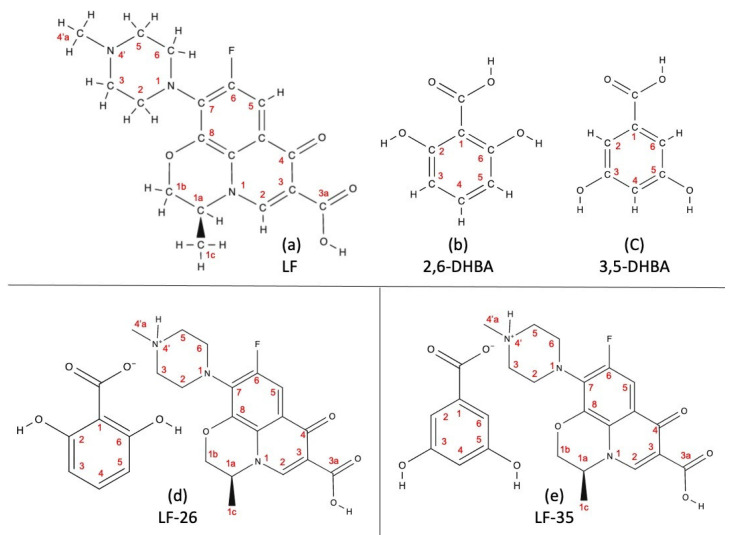
The numbering system for the molecule structures of (**a**) LF; (**b**) 2,6-DHBA; (**c**) 3,5-DHBA; (**d**) LF-26; (**e**) LF-35. Note: LF = levofloxacin; DHBA = dihydroxybenzoic acid; LF-26 = levofloxacin 2,6 dihydroxybenzoate; LF-3,5 = levofloxacin 3,5 dihydroxybenzoate.

**Figure 8 pharmaceutics-15-00124-f008:**
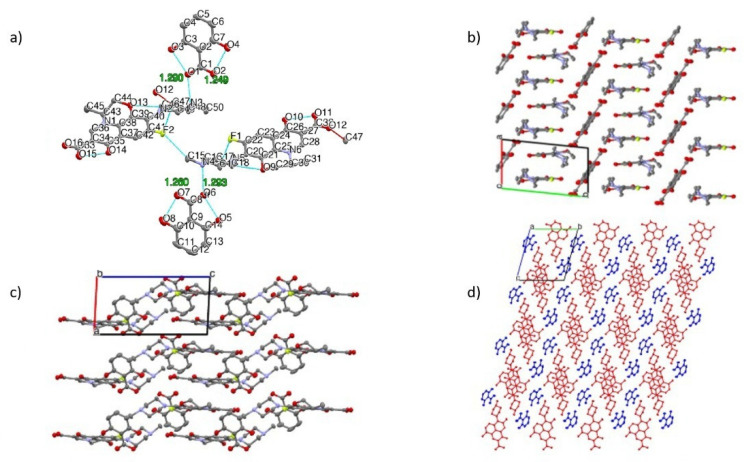
(**a**) Thermal ellipsoid drawing with atomic labeling scheme of LF-26 at 50% probability level. The dashed blue lines indicate intramolecular hydrogen bonds. (**b**) Packing motifs of LF-2,6 along the *c*-axis, hydrogen atoms are omitted for clarity. (**c**) Packing motifs of LF-26 along the *b*-axis and (**d**) *c*-axis with symmetry coloring. The red indicates LF molecules, while the blue color indicates 2,6-DHBA molecules.

**Figure 9 pharmaceutics-15-00124-f009:**
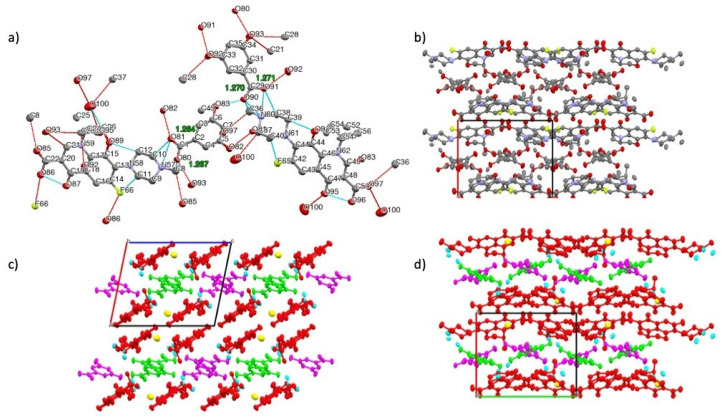
(**a**) Thermal ellipsoid drawing with atomic labeling scheme of LF-35 at 50% probability level. The dashed blue lines indicate intramolecular hydrogen bonds, while the dashed cyan lines indicate intermolecular hydrogen bonds. (**b**) Packing motifs of LF-35 along the *a*-axis, hydrogen atoms are omitted for clarity. (**c**) Packing motifs of LF-35 along the *b*-axis and (**d**) *a*-axis with symmetry coloring. The red color indicates LF molecules, while the magenta and green color indicate 3,5-DHBA molecules.

**Figure 10 pharmaceutics-15-00124-f010:**
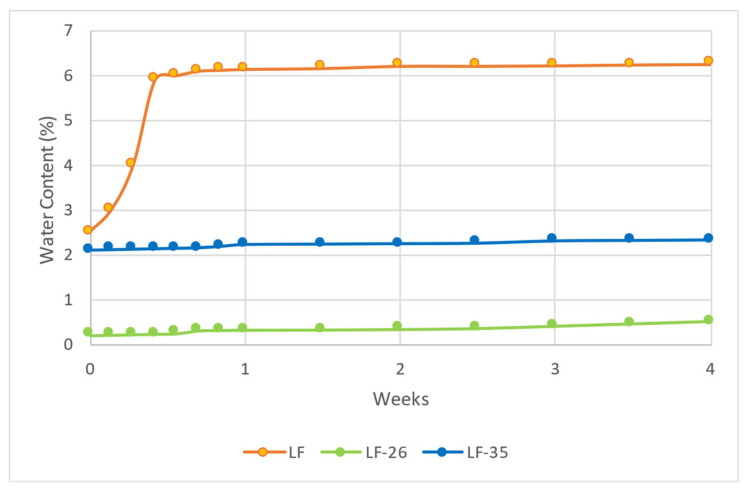
Hydrate transformation profile of LF (levofloxacin), LF-26 (levofloxacin-2,6 dihydroxybenzoate), and LF-35 (levofloxacin-3,5 dihydroxybenzoate) for four weeks in condition 75 ± 1.0% RH/25 ± 0.5 °C.

**Figure 11 pharmaceutics-15-00124-f011:**
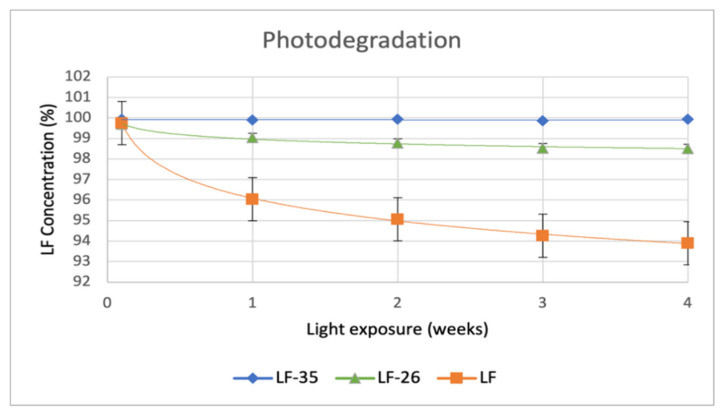
Photodegradation graph of LF, LF-26, and LF-35 for the 4-week stability test period. Note: LF = levofloxacin, LF-26 = levofloxacin 2,6 dihydroxybenzoate, LF-35 = levofloxacin 3,5 dihydroxybenzoate.

**Figure 12 pharmaceutics-15-00124-f012:**
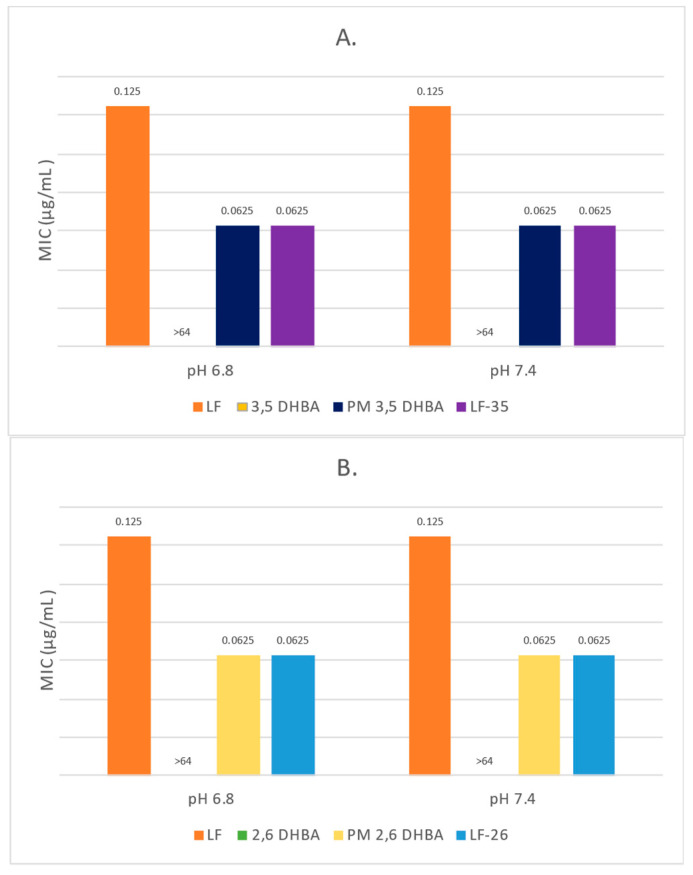
MIC comparison against *Staphylococcus aureus* (**A**,**B**) and *Escherichia coli* (**C**,**D**). Note: LF = levofloxacin, 3,5 DHBA = 3,5 dihydroxybenzoic acid, 2,6 DHBA = 2,6 dihydroxybenzoic acid, LF-26 = levofloxacin 2,6 dihydroxybenzoate, LF-35 = levofloxacin 3,5 dihydroxybenzoate, PM = physical mixture.

**Table 1 pharmaceutics-15-00124-t001:** Crystallographic data and refinement details of LF-2,6-DHBA and LF-3,5-DHBA.

Parameter	LF-26	LF-35·0.5H_2_O
Compound name	Levofloxacin 2.6-dihydroxybenzoate	Levofloxacin 3.5-dihydroxybenzoate hemihydrate
Moiety formula	C_18_H_20_FN_3_O_4_·C_7_H_6_O_4_	C_25_H_26.5_FN_3_O_8.25_
Formula weight	525	519.5
Crystal system	Triclinic	Monoclinic
Space group	*P*1	*P*2_1_
*a* (Å)	6.9081 (1)	11.8145 (2)
*b* (Å)	12.6342 (2)	13.9434 (3)
*c* (Å)	13.9348 (1)	14.5712 (3)
*α* (°)	104.886 (1)	90
*β* (°)	91.446 (1)	101.5950 (10)
*γ* (°)	95.242 (1)	90
Volume (Å^3^)	1168.97	2351.41
*Z, Z’*	2, 2	4, 4
*T* (K)	93	93
R-factor (%)	4.13	4.89
CCDC Deposition number	2180214	2180221

Note: LF-26 = levofloxacin 2,6 dihydroxybenzoate; LF-35 = levofloxacin 3,5 dihydroxybenzoate.

**Table 2 pharmaceutics-15-00124-t002:** Sample appearances during the test of stability towards UV lighting.

Sample Name	Original	Week 1	Week 2	Week 3	Week 4
LF	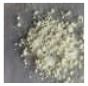	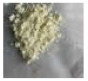	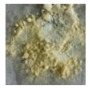	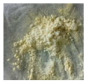	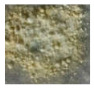
LF-26	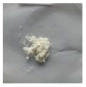	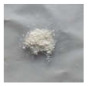	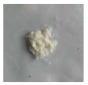	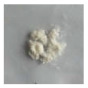	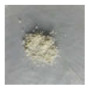
LF-35	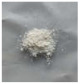	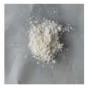	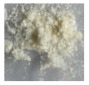	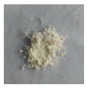	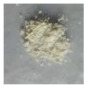

Note: LF = levofloxacin; LF-26 = levofloxacin 2,6 dihydroxybenzoate, LF-35 = levofloxacin 3,5 dihydroxybenzoate.

**Table 3 pharmaceutics-15-00124-t003:** Levofloxacin concentration in the original/untreated samples.

Sample Name	Conc. Value (PPM)	Found Value of LF (PPM)	Concentration Percentage	Mean Concentration	RSD (*n =* 3)
LF	400	399.8	99.9	99.8%	0.3%
		400.1	100.0%		
		397.9	99.5%		
LF-26	200	198.9	99.4%	99.8%	0.36%
		199.4	99.7%		
		200.3	100.1%		
LF-35	600	599.2	99.9%	99.9%	0.16%
		598.8	99.8%		
		600.6	100.1%		

Note: LF = levofloxacin; LF-26 = levofloxacin 2,6 dihydroxybenzoate, LF-35= levofloxacin 3,5 dihydroxybenzoate, RSD = relative standard deviation.

**Table 4 pharmaceutics-15-00124-t004:** Levofloxacin levels after four weeks of testing for stability towards UV lighting.

Sample Name	Conc. Value (PPM)	Found Value of LF (PPM)	Concentration Percentage	Mean Concentration	RSD (*n* = 3)
LF	400	375.2	93.8%	93.9%	0.21%
		376.4	94.1%		
		374.9	93.7%		
LF-26	200	197.5	98.8%	98.5%	0.26%
		196.8	98.4%		
		196.5	98.3%		
LF-35	600	600.1	100.0%	99.9%	0.12%
		598.8	99.8%		
		599.9	100.0%		

Note: LF = levofloxacin; LF-26 = levofloxacin 2,6 dihydroxybenzoate, LF-35 = levofloxacin 3,5 dihydroxybenzoate.

**Table 5 pharmaceutics-15-00124-t005:** Minimum inhibition concentration (MIC) of LF-26 and LF-35 compared to constituents.

MIC (μg/mL) (*n* = 3)								
Sample Name	*S. aureus* (~10^5^ CFU/mL)				*E. coli* (~10^5^ CFU/mL)			
	In pH 6.8 Medium	Final pH	In pH 7.4 Medium	Final pH	In pH 6.8 Medium	Final pH	In pH 7.4 Medium	Final pH
LF	0.125	6.74 ± 0.02	0.125	7.29 ± 0.08	0.125	6.65 ± 0.05	0.125	7.35 ± 0.04
2.6-DHBA	>64	6.45 ± 0.05	>64	7.12 ± 0.03	>64	6.55 ± 0.001	>64	7.32 ± 0.05
3.5-DHBA	>64	6.52 ± 0.01	>64	7.25 ± 0.007	>64	6.75 ± 0.03	>64	7.27 ± 0.06
LF-26	0.0625	6.58 ± 0.03	0.0625	7.25 ± 0.007	0.0625	6.65 ± 0.04	0.0625	7.31 ± 0.002
PM LF-26	0.0625	6.64 ± 0.02	0.0625	7.25 ± 0.002	0.0625	6.67 ± 0.07	0.0625	7.30 ± 0.01
LF-35	0.0625	6.68 ± 0.01	0.0625	7.35 ± 0.03	0.0625	6.72 ± 0.04	0.0625	7.37 ± 0.3
PM LF-35	0.0625	6.75 ± 0.02	0.0625	7.35 ± 0.01	0.0625	6.68 ± 0.02	0.0625	7.33 ± 0.01

Note: LF = levofloxacin; 3,5, DHBA = 3,5 dihydroxybenzoic acid, 2,6 DHBA = 2,6 dihydroxybenzoic acid, LF-26 = levofloxacin 2,6 dihydroxybenzoate, LF-35 = levofloxacin 3,5 dihydroxybenzoate, PM = physical mixture.

**Table 6 pharmaceutics-15-00124-t006:** Fractional inhibitory concentration index (FICI) for the combination of LF with 2,6 DHBA and 3,5 DHBA against *Staphylococcus aureus* and *Escherichia coli*.

Sample Name	*S. aureus* (*n* = 3)							
	In pH 6.8				In pH 7.4			
	FICI of DHBA	FICI of LF	ΣFICI	Interpretation	FICI of DHBA	FICI of LF	ΣFICI	Interpretation
2.6 DHBA	0.062	0.395	0.457	S	0.062	0.395	0.457	S
3.5 DHBA	0.062	0.395	0.457	S	0.062	0.395	0.457	S
Sample Name	*E. coli* (*n* = 3)							
	In pH 6.8				In pH 7.4			
	FICI of DHBA	FICI of LF	ΣFIC	Interpretation	FICI of DHBA	FICI of LF	ΣFICI	Interpretation
2.6 DHBA	0.062	0.395	0.457	S	0.062	0.395	0.457	S
3.5 DHBA	0.062	0.395	0.457	S	0.062	0.395	0.457	S

Note: (S = synergy, A = antagonist, I = indifferent).

## Data Availability

Not applicable.

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
