# Peer review of "Stability and Antibiotic Potency Improvement of Levofloxacin by Producing New Salts with 2,6- and 3,5-Dihydroxybenzoic Acid and Their Comprehensive Structural Study"

_pharmaceutics, 2022, doi:10.3390/pharmaceutics15010124_

Round 1
Reviewer 1 Report (New Reviewer)
The paper describes the physical, chemical, and antimicrobial properties of compounds obtained by combining levofloxacin with 2,6- and 3,5-dihydroxybenzoic.
The topic is interesting and the study is well-designed. However, for my competence, there is a severe flaw in the MIC assays. In fact, the antimicrobial activity studies were not carried out properly, as too high a concentration of bacteria was used. Please also see the comments to lines 261-263 and 284-288 below. The Authors should repeat those assays.
The language should be revised as many sentences are not clear.
Following there are specific comments.
Lines 44-47: Levofloxacin has a broader spectrum with respect to ciprofloxacin, but it is not correct to ascertain that it is more effective against Gram-positive rather than Gram negative bacteria. Neither the cited reference (Aldred et al., 2014) nor the medline records assess that. Levofloxacin is used for treating Gram positive and Gram negative infections. It is just recommended also for a wide range of infections.
Lines 48-50: This sentence is unclear, please rewrite it.
Line 53: “It is not only in an aqueous solution”: This sentence can be removed.
Lines 57-58: Please rewrite this sentence: levofloxacin cannot show bacteria resistance, rather bacteria show resistance to levofloxacin. Also, in the previous sentence levofloxacin is cited, but THE Authors use “they” in this sentence, instead. That makes the statement quite unclear.
Lines 59-60: The syntax of this sentence is incorrect. “Composed” does not appear the best lexical choice. Please note that hereafter no more linguistical suggestions will be provided, the while manuscript needs a deep linguistic revision.
Line 66: (Please write, at its first occurrence, DHBA parenthetically.
Lines 72-73: Please provide a reference for such a statement.
Lines 72-73: Inflammation is not due to gastrointestinal perforation, at least it is the contrary. In any way, the cited reference does not associate inflammation and gastrointestinal perforation.
Lines 65-67 and 76-77: It is not clear why 2,6-DHBA and 3,5-DHBA have been selected. It looks like the Authors performed some preliminary experiments, but no further detail has been provided.
Line 113: Please provide information about the bacterial strains. Where and how were they isolated? What are their antimicrobial susceptibility profile? How were they stored? How were they revitalized?
Line 192: Please indicate the software provider.
Line 218: What is “aquadest”?
Lines 261-263 and 281-284: According to the CLSI standard procedure, the microdilution method requires a final concentration of 5E05 CFU/mL in each well. The Authors obtained a final concentration of 1.5E07 CFU/mL. Consequently, the MIC results cannot be considered.
Lines 263-264: It is not necessary to provide the abbreviated form of the genus at its first occurrence because it is universally accepted.
Lines 268-272 and 284-288: Considering the high concentration in each well, no evaluation was possible, since even the starting bacterial suspension should have been turbid.
Line 307 and Figure 2: Usually, when talking about crystal shape, “cubic” is used more than “square”. However, the cubic shape is not appreciable in Figure 2b, which is too small and out of focus.
Lines 313-315: It is not clear the reason why LF-26 and LF-35 could not be expressed in full, since they are then expanded.
Lines 324-325: The exothermic peak is visible only in LF-26 and LF-35 profiles.
Figure 3: temperature scale should be provided on the x-axis.
Lines 561-564: Actually, the MIC testing at pH 1.2 is quite useless in this case, as the potency is not tested against acidophilic species.
Lines 567-575, Table 6, and Figure 12: Considering the flaw in the setup of MIC tests, it is not clear how the Authors could have determined those values. Please also see the comment on lines 261-263 and 281-284 and the general comments above.
Lines 619-630: Actually, no experimental data have been presented to ascertain the protective or inflammatory inhibition activities of LF-26 and LF-35, therefore this part of the discussion is merely speculative. I would suggest the Authors to remove, or drastically reduce such a part.
Line 641: Talking about antimicrobial potency is speculative, considering that MIC tests have not been properly carried out.
Line 644: The reference to bacterial resistance should be absolutely avoided since no test has been carried out on resistant strains.
Author Response
It is attached. Thank you

Reviewer 2 Report (New Reviewer)
Stability and antibiotic potency improvement of levofloxacin by producing the new salts with 2, 6 – and 3, 5 – dihydroxybenzoic acid and their comprehensive structural study
In the manuscript titled and submitted as “Stability and antibiotic potency improvement of levofloxacin by producing the new salts with 2, 6 – and 3, 5 – dihydroxybenzoic acid and their comprehensive structural study“, the authors have studied the development of two newly formed salts that can be used in the place of levofloxacin which is not stable and highly affected by lighting resulting in phase change and dose. Antioxidant dihydroxydenzoic acid isomers were used to study the formed antioxidants with levofloxacin namely 2, 6 – dihydroxybenzoic acid and 3, 5 – dihydroxybenzoic acid. They were characterized using thermal analysis, neutron magnetic resonance, Fourier transform infrared spectroscopy and powder X-ray diffractometry. The formed antibiotics possessed different structure, stable and can be good antibiotics suited for solid and liquid dosages.
Methodology is clearly explained. Tables show a clear comparison between the varieties and the images are nice. References are new and relevant.
Major points:
1. The stability test was done to simulate the environment in Indonesia. Is it prepared only to satisfy this environment or can it be exported to other countries. If so, is there any other regulatory measure needs to be fulfilled.
2. In Figure 2, Single crystal appearance the newly formed states namely 2, 6 – dihydroxybenzoic acid and 3, 5 – dihydroxybenzoic acid were shown in two different magnifications (4 & 40). Why? Same magnification may be easier for comparison.
3. In that, needle shape is clear whereas the other structure is not square as the one at the bottom and other at the right seems to be different.
4. Two salts were successfully prepared and reported. How do they behave under other pH environment? Will they be stable?
Minor points:
1. A single sentence depicting difference between gram positive and gram negative bacteria may be added.
2. It was reported in the introduction that DHBA protects the gastrointestinal disorders. Kindly include references.
3. Kindly expand pKa, ICH, Q1B, CLSI, ABTS, FRAP, CUPRAC, RSD and cfu.
4. The first sentence of the conclusion seems to be truncated.
The authors have done an exploratory work on stability and antibiotic potency improvement of levofloxacin and the paper is suitable for publication in the journal “Pharmaceutics” following revision.
Round 2
Reviewer 1 Report (New Reviewer)
I find the Authors significantly improved the manuscript. I would just like to bring to their attention that the sentence in lines 68-69 is still not correct: "Moreover, these antibiotics also show signs of bacteria resistance since they have been used for a long time". Antibiotics are inanimated, they cannot show resistance. Bacteria show resistance to an antibiotic.
Author Response
Please kindly find the attachment. Thank you.

This manuscript is a resubmission of an earlier submission. The following is a list of the peer review reports and author responses from that submission.
Round 1
Reviewer 1 Report
Nugrahani et al. have proposed two Levofloxacin salt variants and identified/evaluated their physicochemical properties. Although the findings described in this paper have some academic importance, these salts would not be developed as new active pharmaceutical ingredients (APIs) based solely on slight improvements in solubility, stability, and efficacy. To be accepted by a journal at the level of Pharmaceutics requires more interesting findings, such as dramatic improvements in these properties would be needed. The authors have previously published similar articles describing complexes/salts of Levofloxacin, and the experimental difficulty of the methods described in this paper appears to be quite low. At a molar ratio of 1:1, a complex formation would be easily expected. The novelty and/or necessity of this study are insufficient to attract readers of this journal. Some figures are unclear, and typos are observed throughout the manuscript. Based on these observations, this paper is not suitable to be published in Pharmaceutics.
Author Response
Dear Reviewer,
Please see attachment for the point-by-point response.
Thank you for your time.
Best regards,
Ilma Nugrahani

Reviewer 2 Report
In this study Authors have present how Levofloxacin (LF) a fluoroquinolone antibiotic is sparingly soluble 9 in water and degrades due to light and moisture. This results in limiting its therapeutic development. Authors state an describe in detail how these characteristics can be improved by composing a multicomponent system of LF with dihydroxybenzoic acids (DHBA).
This study is very interesting, and Authors have clearly described all the experimental details very precisely. Before suggesting this for editorial decision, I need some clarification about Antimicrobial activity study. Table 6 can be more helpful for readers to follow easily if MIC are represented in the form of a Figure instead of table (pH 1.2 data is fine within the embedded text). Presenting the rest of the table as figure will be a good addition.
Also Authors need to describe number of technical or biological replicates used for MIC calculation. This information along with standard deviation should be clearly described in the Figure legend.
Author Response

(The authors gave the same response as above.)

Reviewer 3 Report
1. Abstract section, to avoid high similarity index the author have used certain words that are clinically incorrect. For example in the first sentence, wide spectrum, limiting its development.
2. The whole abstract section needs to be revised with a sequence of method, result and conclusion. In the preset form it is merely a piece of discussion.
3. Arrange keywords list alphabetically and omit those words that are part of manuscript title
4. In introduction there is lack of sequence and also relevant literature has not been cited. For example a very study: Levofloxacin cocrystal/salt with phthalimide and caffeic acid as promising solid-state approach to improve antimicrobial efficiency. Antibiotics 2022, 11, 797. https://doi.org/10.3390/antibiotics11060797
Enhancing Dissolution Rate and Antibacterial Efficiency of Azithromycin through Drug-Drug Cocrystals with Paracetamol. Antibiotics 2021, 10, 939. https://doi.org/10.3390/antibiotics10080939 Synthesis of cefixime and azithromycin nano particles; an attempt to enhance its antimicrobial activity and dissolution rate. Journal of Nanomaterials. 2016; 1-9.
5. Also, in introduction the novelty statement is poor and needs to be revised.
6. Provide a rationale about the intervening substances in introduction section.
7. Section 2.2.12 is too long and needs to be revised.
8. Include a section as statistic analysis in methodology section.
9. Figures are ok. Only enhance its resolutions
10. Use uniform style for reference list.
11. Rephrase the conclusion section
Author Response

(The authors gave the same response as above.)

Reviewer 4 Report
I believe the paper deserved to be published in Pharmaceutics once all the following suggestions/comments will be addressed.
In the introduction, some English changes are required:
"Besides that, previously LF has also been combined with other compounds, such as pyrazinamide, carbamazepine, ubiquinol, and piroxicam, which was shown to alter the obtained compound’s properties.”
“This benzoic acid derivate is a naturally occurring compound in nature,…”
In the introduction, the authors should stress better the reasons why the instability in water and to light are problematic for LF referring to LF formulations on the market. Is LF available in solution?
2.1. Materials: please correct “aluminum plate” in “aluminum pan”
2.2. Sample preparation: what did the authors meant with “aqueous methanol-ethanol mixture”? There is water in the mixture?
Line 78: correct “under room temperature”
Line 78: “A small new phase…” what is the meaning of this?
2.2.3. Electrothermal analysis. The analysis description has already been reported in section 2.2.1. Please avoid repetition.
Line 94: “pellet” I do not believe a pellet was obtained but a tablet.
2.2.10. Why did the authors investigate the solubility in water? I do not believe it is a medium of reference for the determination of solubility. On the contrary, the authors had to evaluate the solubility in three different solutions in the pH range 1.2-6.8. A buffer at pH 4.5 had to be used for the solubility study in agreement with the WHO protocol and the ADMET white paper (doi: 10.5599/admet.4.2.292). In addition, references lack is this section and the temperature at which the study was performed is not reported.
Lines 150-151: “continued until solid residue remained” What was the duration of the study? 24 hours? 48-72 hours? Time is of paramount importance in determining the solubility. Are the authors referring to equilibrium solubility? It would be of interest to determine the intrinsic solubility. Then, the solid residue should be recovered and characterize to determine its nature.
2.2.11. What type of climatic chamber did the authors employed?
2.2.12. Avoid repetitions. The protocol for the preparation of buffer pH 1.2 has already been described. Why did the authors prepared a buffer pH 6.8 with salts different from those used to prepare buffer pH 6.8 of the solubility study?
Line 229: “water release” The terms employed are not suitable.
Lines 421-422: LF-35 is not characterized by a lower melting temperature with respect to LF-26. The author statement makes no sense.
Table 3, line 463 and line 491: organoleptic means taste, color, odor, and feel of a substance. The authors only examine the appearance of the substance. Please correct.
Lines 460-461: Please correct the units that are in superscript.
Line 498: why the authors have not investigated the hygroscopicity.
Author Response

(The authors gave the same response as above.)

Round 2
Reviewer 1 Report
The author's response did not affect my first opinion "although the findings described in this paper have some academic importance, this paper is not suitable to be published in Pharmaceutics."
Reviewer 2 Report
Authors have addressed all my comments and added necessary information in the article. I strongly recommend this for publication in present form.
Reviewer 3 Report
ok
Reviewer 4 Report
English needs further improvements. For example: There are several singular/plural improper uses. Why the authors refer to “another antioxidant”? Another with respect to ….?
A solution to the stability problem could be more simply the extemporaneous preparation of the syrup, suspension and solution for injection. I do not believe LF instability in water is a big concern. What about the exposure to light?
I believe the method section is not well structured. 13 subsections are too much, they could be easily combined where appropriate.
Correction 9: Line 94: “pellet” I do not believe a pellet was obtained but a tablet.
Response 9: We have replaced “pellet” with “a thin and transparent tablet” in Line 507 as follows. Thank you.
“…compressed into a thin and transparent tablet, and put in the sample holder.”
Tablet is enough, there is no need to specify its appearance, this is common knowledge.
Why did the authors used 25 °C for their study?
How the authors define hygroscopicity? Is LF hygroscopic? I do not believe the study was carried out appropriately. There was no control of temperature and relative humidity. Then, 5% water absorption in not significant. Conclusions must be edited, salts were not discovered but prepared by the authors. It is not clear from the conclusions what are the advancements of this research. I believe LF instability is not a problem and LF is reported to be soluble to freely soluble so it is not clear what is the advantage of increasing its solubility up to 4-folds.
@font-face {font-family:"Cambria Math"; panose-1:2 4 5 3 5 4 6 3 2 4; mso-font-charset:0; mso-generic-font-family:roman; mso-font-pitch:variable; mso-font-signature:3 0 0 0 1 0;}@font-face {font-family:Calibri; panose-1:2 15 5 2 2 2 4 3 2 4; mso-font-charset:0; mso-generic-font-family:swiss; mso-font-pitch:variable; mso-font-signature:-536859905 -1073732485 9 0 511 0;}@font-face {font-family:"Palatino Linotype"; panose-1:2 4 5 2 5 5 5 3 3 4; mso-font-charset:0; mso-generic-font-family:roman; mso-font-pitch:variable; mso-font-signature:-536870265 1073741843 0 0 415 0;}p.MsoNormal, li.MsoNormal, div.MsoNormal {mso-style-unhide:no; mso-style-qformat:yes; mso-style-parent:""; margin-top:0cm; margin-right:0cm; margin-bottom:8.0pt; margin-left:0cm; line-height:107%; mso-pagination:widow-orphan; font-size:11.0pt; font-family:"Calibri",sans-serif; mso-ascii-font-family:Calibri; mso-ascii-theme-font:minor-latin; mso-fareast-font-family:Calibri; mso-fareast-theme-font:minor-latin; mso-hansi-font-family:Calibri; mso-hansi-theme-font:minor-latin; mso-bidi-font-family:"Times New Roman"; mso-bidi-theme-font:minor-bidi; mso-ansi-language:EN-ID; mso-fareast-language:EN-US;}.MsoChpDefault {mso-style-type:export-only; mso-default-props:yes; font-size:11.0pt; mso-ansi-font-size:11.0pt; mso-bidi-font-size:11.0pt; font-family:"Calibri",sans-serif; mso-ascii-font-family:Calibri; mso-ascii-theme-font:minor-latin; mso-fareast-font-family:Calibri; mso-fareast-theme-font:minor-latin; mso-hansi-font-family:Calibri; mso-hansi-theme-font:minor-latin; mso-bidi-font-family:"Times New Roman"; mso-bidi-theme-font:minor-bidi; mso-ansi-language:EN-ID; mso-fareast-language:EN-US;}.MsoPapDefault {mso-style-type:export-only; margin-bottom:8.0pt; line-height:107%;}div.WordSection1 {page:WordSection1;}